# The Association of Social Support and Symptomatic Remission among Community-Dwelling Schizophrenia Patients: A Cross-Sectional Study

**DOI:** 10.3390/ijerph18083977

**Published:** 2021-04-09

**Authors:** Chi-Hsuan Fan, Shih-Chieh Hsu, Fei-Hsiu Hsiao, Chia-Ming Chang, Chia-Yih Liu, Yu-Ming Lai, Yu-Ting Chen

**Affiliations:** 1Department of Nursing, Chang Gung Memorial Hospital, Linkou, Taoyuan City 333, Taiwan; jun3456x2@cgmh.org.tw; 2Department of Psychiatry, Chang Gung Memorial Hospital, Linkou, Taoyuan City 333, Taiwan; hsu3160@cgmh.org.tw (S.-C.H.); cmchang58@yahoo.com.tw (C.-M.C.); liucy752@cgmh.org.tw (C.-Y.L.); 3Department of Psychiatry, New Taipei Municipal TuCheng Hospital, New Taipei 236, Taiwan; 4College of Medicine, Chang Gung University, Taoyuan City 333, Taiwan; 5School of Nursing, College of Medicine, National Taiwan University, Taipei City 100, Taiwan; hsiaofei@ntu.edu.tw; 6Department of Nursing, National Taiwan University Hospital, Taipei City 100, Taiwan; 7School of Nursing, College of Medicine, Chang Gung University, Taoyuan City 333, Taiwan; yulai@mail.cgu.edu.tw

**Keywords:** schizophrenia, symptomatic remission, social support, family support, community, Taiwan

## Abstract

Schizophrenia is a mental disease that often leads to chronicity. Social support could reduce the severity of psychotic symptoms; therefore, its influence on remission should be examined. This study investigated the remission rates in community-dwelling schizophrenia patients and examined the association between social support and remission status. A cross-sectional study was conducted in 129 schizophrenia patients in Taiwan. Remission rates were evaluated, and the level of social support, clinical characteristics, sociodemographic variables, and healthy lifestyle status were compared between the remission and nonremission groups. The association between social support and remission was analyzed after adjusting for confounding factors. The mean illness duration is 12.9 years. More than 95% of the participants lived with their families, 63% were unemployed, and 43% achieved remission. Higher social support was observed in the remission group, and a significant correlation was observed between family domain of social support and remission status. Family support was a protective factor of symptomatic remission in community-dwelling schizophrenia patients in Taiwan. The results reflect the effects of a family-centered culture on patients during illness. Consequently, reinforcing family relationships and the capacity of families to manage the symptoms of patients and providing support to families are recommended.

## 1. Introduction

Schizophrenia is a severe and chronic mental disorder that affects more than 21 million individuals globally, with a lifetime prevalence of 1% and 0.87% globally and in European countries, respectively [1,2]. According to the Ministry of Health and Welfare, Taiwan, in 2019, approximately 11% of the individuals in Taiwan who received Welfare for People with Disabilities from the government were in chronic mental health conditions [3]. The rehospitalization rates of schizophrenia patients in Taiwan within 1 and 5 years of discharge from hospitals have been reported to be 22.3% and 37.8%, respectively [4]. Poor remission of symptoms in schizophrenia patients and high rehospitalization rates not only represent economic burdens to the families of patients but also increase mental health care costs for caregivers [5,6]. Consequently, the study of remission in schizophrenia patients and factors that influence it are critical and have been the focus of numerous studies.

Symptom remission and functional recovery are considered prognostic indicators in patients with schizophrenia [7]. Symptom remission is a prerequisite for recovery and has been increasingly highlighted as an indicator of prognosis because it has been well-defined [7,8]. The Remission in Schizophrenia Working Group (RSWG) has identified different instruments according to the stage of schizophrenia for the definition of remission [8], including Brief Psychiatric Rating Scale, Scale for the Assessment of Positive Symptoms, Scale for the Assessment of Negative Symptoms [9], and Positive and Negative Syndrome Scale (PANSS) [10]. The definition of remission proposed by the RSWG [8] is widely recognized and applied in research across various countries [11,12]. It has been suggested that simpler tools are required to assess symptom remission in clinical settings [13], such as Clinical Global Impression (CGI) [14]. However, considering that the scale can only provide a general assessment based on only two questions, it may not be an objective assessment for major symptoms, as in the case of PANSS.

Schizophrenia remission rates vary largely based on the study design and target population. A review study reported that 17–78% and 16–62% of first-episode and multiple-episode schizophrenia patients achieved full remission criteria (remission within six months), and no significant difference was reported between the remission rates of the two groups [15]. In addition, a study reported a remission rate of 63% for schizophrenia patients at discharge after inpatient treatment and 45.1–55.1% for patients within 12 months of an acute episode [16,17]. Conversely, a cross-sectional study reported a 31–38% remission rate in schizophrenia patients in communities in European countries [18]. Another study on community-dwelling older schizophrenia patients reported 23.4% remission rates at baseline, and almost 86% of the patients maintained the remission status at five years; however, among 72.7% of patients who were not in remission, only 35.6% had achieved remission at 5 years [19]. Some longitudinal studies have reported 52–71% remission rates for patients with acute schizophrenia episodes and who had been discharged from hospital within a year [20,21,22], whereas other studies that followed up illness progression from 6 to 11 years reported 17.2–37% remission rates [23,24]. In a study in Taiwan, the remission rates in schizophrenia patients was 36.7% 1 month after discharge [25]. Lang et al. reviewed 21 studies and revealed a sustained remission rate of 52.6% after 6 months in some studies [26]. Most of the aforementioned findings reveal that sustained remission could be enhanced in schizophrenia patients in communities and highlights the importance of exploring factors that are associated with the remission in such populations.

Social support is a multidimensional concept, and has been previously defined as interpersonal relationships that influence the psychological and social functioning of individuals [27]. According to Cohen and Wills, social support can be divided into esteem support, informational support, social companionship, and instrumental support [28]. Social support is also defined as the amount of assistance from the interpersonal networks of an individual that can improve personal well-being, decrease uncertainty, increase control over life, and enhance mental health [29]. In addition, according to Dambi et al., support can be broadly classified into emotional, practical, or informational support [30]. Social support is indispensable for schizophrenia patients and can reduce the severity of psychotic symptoms [31]. In addition, it can facilitate adaptation to adverse circumstances within a community or recovery from the disease, and it can enhance quality of life [32,33]. Social support has been found to be a protective factor for social functioning and quality of life [32,34], and perceived better social support showed a mediate effect on the relation between social interaction and psychotic symptoms [35]. Since remission status is a prognostic indicator as mentioned, exploring the impact of different dimensions of social support on symptom remission is critical. In a previous study, early social support, one year after treatment, predicted remission in negative symptoms and functional recovery at five years in patients with psychotic disorders, and social support in the study was assessed by the case managers of patients based on three items from the Wisconsin Quality of Life Scale-Provider Version [36]. Some studies have focused on the relationship between social function and remission status [23,37,38]. Other studies have found that schizophrenia patients with higher frequencies of social interaction are more likely to achieve symptomatic remission [39,40]. Brissos et al. reported that patients with better interpersonal and social relationships are more likely to achieve remission [18]. A meta-analysis study concluded that schizophrenia patients with smaller social network size suffered from more severe psychiatric symptoms [41]. Since social interaction frequency and interpersonal and social relationships from the studies above may not be able to fully reflect a multiple dimension of social support. More indicators with a complete concept of social support are required to validate its impacts on symptomatic remission.

With regard to the association between sociodemographic characteristics and remission status, although some studies have reported higher rates of remission in female patients than in male patients [21], other studies have reported inconsistent findings [23,24,40,42]. In addition, higher levels of education are associated with higher levels of remission [25,39,40], and significant differences have been observed in employment status remission and nonremission groups [11]. Schennach et al. also reported that 69.23% of patients in the remission group were employed, which was significantly higher than the proportion in the nonremission group (30.77%) [7]. However, Jaracz et al. observed no correlation between the capacity to work and symptomatic remission [23].

Living a healthy lifestyle may also influence remission in schizophrenia patients. Patients with schizophrenia are at higher risks of cardiovascular disease, metabolic syndrome, and obesity, and they are more likely to be heavy smokers and are less likely to engage in exercise, which would all decrease average life expectancy than that in the general public [43,44]. Exercise improves health and cognitive function, which in turn influences memory and attention, in schizophrenia patients [45,46,47]. Smoking is also very common among individuals with mental disorders, particularly among male and younger individuals [48]. The smoking rate in schizophrenia patients is 2–3-fold of the rate in the general population [49]. A study exploring the association between smoking and psychiatric symptoms in 367 patients with schizophrenia reported that the smoking group had significantly more severe psychiatric symptoms than the nonsmoking group [50]. In addition, adherence to antipsychotic medication indirectly influences symptom resolution [51].

Shorter duration of untreated psychosis (DUP) is a predictor of remission in first-episode schizophrenia patients [21,52]. DUP also influences the remission rate of negative symptoms [53]. Although the duration of illness is associated with remission, the correlation is inconsistent across different studies [7,18,24,39]. According to Helldin et al., patients with shorter hospital stay in the previous year are more likely to achieve remission [39], although inconsistent findings have been reported [18,40]. Regarding antipsychotic medication treatment, schizophrenia patients treated with atypical antipsychotics are more likely to achieve remission compared with patients treated with typical antipsychotics [54], and patients receiving antipsychotic long-acting injections have significantly higher remission rates than patients receiving oral antipsychotics [55].

Based on the aforementioned literature review, studies on symptomatic remission in Asian countries are scarce, particularly on schizophrenia outpatients. Previous studies on the relationship between social support and remission have mainly focused on the effects of social interaction frequency or interpersonal relationship; therefore, more valid instrument are required for the assessment of the influence of social support on remission status. Consequently, the objective of the present study was to (a) investigate the status quo regarding symptomatic remission in Taiwanese patients with schizophrenia and (b) investigate the association between social support and remission status in schizophrenia patients in Taiwan after controlling for confounders. We hypothesize that community-dwelling schizophrenia patients with better social support are more likely to achieve symptomatic remission.

## 2. Materials & Methods

### 2.1. Study Setting and Design

A cross-sectional study was conducted between March and September 2018. Patients were recruited from a psychiatric outpatient department (OPD) and a psychiatric day care center at a medical center in northern Taiwan.

### 2.2. Participants

The inclusion criteria for participants were as follows: (1) schizophrenia diagnosed by a psychiatrist and (2) age between 20 and 80 years. The patients who were in unstable condition such as violent or suicidal, and the patients who had organic brain disease, mental retardation, substance abuse, or whose cognitive impairment was severe enough to negatively affect the authenticity of the study data, were excluded. 

### 2.3. Measures

#### 2.3.1. Symptomatic Remission 

We used Positive and Negative Syndrome Scale-8 (PANSS-8) items plus Clinical Global Impression-improvement (CGI-I) to assess symptomatic remission. Andreasen et al. [8] further developed a short version of the original PANSS scale [56], consisting of the following eight items: P1, Delusion; P2, Conceptual Disorganization; P3, Hallucinatory Behavior; N1, Blunted Affect; N4, Passive/Apathetic Social Withdrawal; N6, Lock of Spontaneity/Flow of Conversation; G5, Mannerisms and Posturing; and G9, Unusual Thought Content. Symptom severity was evaluated on a scale of 1 (none) to 7 (overwhelming). The scale has been translated into multiple languages and is used extensively [11,40,53]. The PANSS-8 scale has been tested and reported to exhibit acceptable internal consistency and high correlation with PANSS-30 [57]. The Taiwanese version of the PANSS-30 was translated in 1995, and it exhibits high reliability and validity [58,59]. The Cronbach’s α for PANSS-8 in the present study was 0.81, indicating good internal consistency. Because of the limitations of the cross-sectional study design, we added the following question in the CGI-Improvement scale (CGI-I), “In your opinion, how much has this patient changed within 6 months since the start of treatment?”, to assess change in condition within 6 months [13,60]; it was scored from 1 (much improved) to 7 (much worse). The two scales were both assessed by psychiatrists. Symptomatic remission was defined as a score of 3 or lower on all PANSS-8 items plus a score of 4 or lower in CGI-improvement.

#### 2.3.2. Social Support

Multidimensional Scale of Perceived Social Support (MSPSS), a 12-item scale developed by Zimet et al. [58], is one of the most widely used instruments for measuring social support [30]. The MSPSS, which is self-rated, measures the perceived adequacy of the available level of social support from three sources: family, friends, and significant others. The scoring is based on a 7-point Likert scale, from 1 (strongly disagree) to 7 (strongly agree). The construct validity of the tree-factor structure has been confirmed, and good internal consistency of the overall scale and the three subscales has been demonstrated, with Cronbach’s α values ranging between 0.85 and 0.91 [61]. The scale has been translated widely into multiple languages. The Chinese version of MSPSS was translated by Wang [62], and its reliability and validity were tested in Asian schizophrenia outpatients [63]. In the present study, the sums of total scale and three subscales were calculated. The Cronbach’s α value of the scale, 0.89, indicated good internal consistency. 

#### 2.3.3. Clinical and Sociodemographic Characteristics and Healthy Lifestyle Status

The clinical characteristics of the disease included the age of onset, the duration of illness (year), the DUP (day) within a year, the number of schizophrenia hospitalizations, and the types of antipsychotics being administered. We obtained the associated data from electronic medical records to save time and avoid potential errors associated with memory in the responses of patients.

Data on sociodemographic characteristics (gender, age, residence type, education level, employment status, and marital status) and healthy lifestyle status were also obtained. Healthy lifestyle status variables included current smoking status, weekly exercise habits, and adherence to antipsychotic medication regimens. Patients who responded “yes” to the question “I regularly take the antipsychotics daily” and “no” to the following three questions: “I sometimes forget to take antipsychotics,” “I sometimes change the dosage of antipsychotics by myself,” and “I sometimes stop taking antipsychotics for a while” were considered to adhere to antipsychotic medication regimens.

### 2.4. Study Procedure and Data Collection

After the approval by Institutional Review Board (IRB), the researcher collaborated with psychiatrists to recruited participants. Patients who met the inclusion criteria at the psychiatric OPD were enrolled into the study using convenience sampling. All procedures and study purposes were explained to the participants, and they completed and signed a consent form before participation in the study. The researcher also mentioned the right, which was addressed in consent form, to withdraw at any point during the study if the participants feel uncomfortable of participating. All participants signed a consent form before participation in the study, and filled out a structured questionnaire composed of MSPSS, sociodemographic characteristics and healthy lifestyle status with the researcher’s assistance item by item. A gift certificate worth US$4 was offered to participants in recognition of the time they had contributed. The PANSS-8, insight in schizophrenia, and CGI-I scales were scored by psychiatrists during patient visits. The researcher then collected clinical characteristics data by reviewing medical records.

### 2.5. Statistical Analysis

We used the MAC version of IBM SPSS Statistics 20 (IBM Corp., Armonk, NY, USA) for data analysis. The patients were divided into two groups based on age: ≤40 and >40 years, as it was the cutoff point that with the least significant difference between the remission and nonremission groups. Age of onset was also divided into three groups: <18 years (early onset), 18–40 years, and >40 years (late onset) [64,65]. In bivariate analyses, the differences in social demographics, healthy lifestyle status, clinical features, and social support between symptomatic remission and nonremission were analyzed using the chi-square test or independent t-test. Subsequently, all the variables with a *p* value of <0.2 were included in the logistic regression analysis as confounders to validate the correlation observed between social support and symptomatic remission. The appropriate sample size was estimated based on the results of Schennach et al. [7]. First, the odds ratio was calculated according to the rates of regular social interaction and partnership both in the remission group and nonremission group. Thereafter, we calculated the sample size for logistic regression using G-Power v3.1.2 (Heinrich-Heine-University, Dusseldorf, Germany), yielding 83–107 subsequently. In addition, we anticipated a 30% dropout rate in the study, so that the final estimated sample size was 115–150.

## 3. Results

The present study was conducted from 16 March 2018 to 7 September 2018. A total of 129 patients with a mean age of 45 years participated in the study, and the mean duration of illness was 12.9 years. Only 17.1% of the patients were married, and more than 95% lived with their families. Almost 63% of the patients were unemployed, and 23.3% were undergoing vocational/occupational rehabilitation. More than 70% of the patients reported annual family incomes < NT $490,000 (approximately US $16,300), which fell within the lowest 30% of the average disposable income in Taiwan in 2018. [66] Of the 129 participants, 56 (43%) achieved symptomatic remission. 

### 3.1. Sociodemographic Characteristics and Healthy Lifestyles in Symptomatic Remission and Nonremission Groups

The bivariate analysis results revealed no significant differences in gender, marital status, education levels, employment status, family annual income, smoking status, weekly exercise habits, and adherence to antipsychotic medication regimens between the remission and nonremission groups (Table 1). Several variables resulted with a *p* value < 0.2 that were to be selected in multivariate analysis, including aged 40 years and less vs. aged above 40 ($x^{2}$ = 2.77, *p* = 0.09), with a family history of mental illness vs. without a family history of mental illness ($x^{2}$ = 3.47, *p* = 0.06), and exercised weekly vs. no ($x^{2}$ = 2.69, *p* = 0.10).

### 3.2. Clinical Characteristics between Symptomatic Remission and Nonremission Groups

The bivariate analyses results revealed no significant difference in first-episode age group, current treatment, previous psychiatric hospitalization, number of hospitalizations, last discharge time, antipsychotic treatment regiment, type of antipsychotic administered, duration of disease, DUP within a year, and number of schizophrenia hospitalizations between patients in the symptomatic remission group and the nonremission group (Table 2).

In addition to assessing symptomatic remission in our study based on PANSS-8 and CGI-I, we examined if the remission criteria under both scales (CGI-I ≤ 4 vs. all PANSS-8 items ≤ 3) were comparable. According to the results, the percentage of patients scoring CGI-I ≤ 4 were higher in the remission group than in the nonremission group based on PANSS criteria, and the difference was close to significant ($x^{2}$ = 3.16, *p* = 0.07). 

### 3.3. Differences between Social Support and Symptomatic Remission between Symptomatic Remission and Nonremission Groups

The overall social support score in the remission group was significantly higher than that in the nonremission group (46.43 ± 15.0 vs. 39.7 ± 15.2, t = −2.51, *p* = 0.01). Further comparison across three domains revealed that the family domain scores in the remission group were significantly higher than the scores in the nonremission group (22.23 ± 6.0 vs. 19.81 ± 6.1, t = −2.26, *p* = 0.03). Similarly, the scores of the friend domain and significant others domain in the remission group were higher than the corresponding scores in the nonremission group; however, no significant difference was observed between the remission and nonremission groups (Figure 1).

### 3.4. Associations between Social Support and Symptomatic Remission

We conducted multiple regression analysis to examine the influence of social support on symptomatic remission. Variables with *p* < 0.2 based on bivariate analysis (age, family mental illness history, and weekly exercise habits) results were considered confounders and were included in the logistic regression model. The variance inflation factors (VIF) were all lower than 10, indicating no collinearity. After controlling for some social demography variables and healthy lifestyle status, the family domain of social support significantly influenced symptomatic remission (Adjusted Odds Ratio = 1.07, 95% CI = 1.01–1.14), so that patients with higher levels of family support were more likely to achieve symptomatic remission (Table 3). 

## 4. Discussion

The present study explored the symptomatic remission status quo in schizophrenia patients in Taiwan based on symptomatic remission criteria recognized by the RSWG. The application of common indicators in the assessment of symptomatic remission enables the comparison of the effectiveness of various treatments across studies.

Considering the cross-sectional design of the present study, the standard of full remission defined by the RSWG was not met. Full remission is achieved if all the scores for each question in the PANSS-8 scale are ≤3 in each assessment within six months. Therefore, we added CGI-I to assess symptom stability within six months. Of the 129 patients enrolled in the present study, 56 (43%) achieved symptomatic remission according to the new definition. Previous studies have reported remission rates in chronic schizophrenia patients ranging from 31% to 38% [18,39]. In Li et al.’s study, the remission rate in patients with schizophrenia after one month of discharge was 37% [25].

The remission rates in our study and the two aforementioned studies vary greatly when compared with the rates in three studies that enrolled participants with acute-phase first-episode schizophrenia, with 60–91.5% remission rates after hospitalization [20,21,22]. In the present study, we enrolled participants with chronic schizophrenia, which is a phase with residual symptoms; therefore, remission was less likely to be achieved for negative symptoms when compared with positive symptoms in the acute phase [24,26]. Conversely, other studies with longitudinal designs adopted the full remission definition of RSWG, which is more rigorous, and symptomatic remission is less likely to be achieved, as a six-month period is required [8]. Even though the remission rates reported under different study designs varied from that observed in our study, the results of the present study might be more typical for community-dwelling patients when compared with the results reported in other cross-sectional studies, because our study had a larger sample size than the sample size adopted in Li et al. study [25], and we mainly enrolled OPD patients in the course of their regular treatments.

The major finding of our study is that schizophrenia patients with higher social support levels experience higher rates of remission, and family support is an indispensable factor for symptomatic remission. In our study, schizophrenia patients had an overall social support score of 42.62 ± 15.4, which is considered poor perceived social support [61], and which is consistent with the findings of another study that targeted the same population [32]. Compared with other studies that reported that individuals with higher social interaction frequency or better interpersonal and social relationships are more likely to achieve remission [18,39,40], our study conducted a more comprehensive and representative assessment of social support. Therefore, the results could reliably reflect the correlation between symptomatic remission and social support. Social support is one of the key factors facilitating adaptation to adverse circumstances and symptomatic remission in schizophrenia patients in communities [31,32,33]. Social support is often derived from intimate relationships and is essential for mental health [22]. The significant association between the family domain of social support and symptomatic remission reveals the key role of family support for schizophrenia patients during illness when compared with other sources of support.

Family members have been reported to comprise the major social network for patients with psychotic disorders [67]. Other studies on schizophrenia patients reveals different findings that support from friends predicted the quality of life [31]; and support from neighbor/community, other than from family, is more likely to reach functional remission [68]. There are two potential reasons for the inconsistency. First, the average duration of disease in the study patients was 12.9 years, and the symptoms remained largely unchanged over the six-month period based on the CGI-I assessments. Therefore, the participants of the present study mostly had chronic schizophrenia, with long courses of disease and chronicity, interpersonal withdrawal, and social interaction impairment. Conversely, most of the patients were unmarried and unemployed; therefore, their family members were the major social support providers. Second, family values are more emphasized in the East than in the West. The participants in our study were Taiwanese. Due to the family-centered culture in Taiwan, they believe that family members should always get together and even live together; on the contrary, in Western countries, people pay more attention to their close circles of friends. In other words, family members are mostly the care givers for patients with schizophrenia in Taiwanese culture until they achieve remission [69]. Therefore, the relationship between families and patients could influence the severity of the symptoms and the course of the disease [70].

We suggest to strengthen families in providing support to their schizophrenia patients. However, even though patients with family caregivers experienced higher rates of remission and family is considered an important source of support for schizophrenia patients during illness [71,72], family caregivers still report feeling burdened following interactions with schizophrenia patients and when taking care of them [72]. In addition, such family caregivers potentially require more support in terms of health education and rehabilitation information [6]. Therefore, besides evaluating the availability of family support, health care providers should also assess family needs, and strengthen their capacity to manage the symptoms of a patient, so that family caregivers can feel empowered, in turn, increasing remission rates. In addition, assisting patients to expand their social networks is critical for bolstering their social support systems, which could be achieved through appropriate interventions that increase their social participation, strengthen their interpersonal networks, and minimize their sense of isolation. In Taiwan, psychiatric OPD at hospitals are one of the main services for community-dwelling schizophrenia patients to follow up the progress of their disease, therefore developing a nurse-guided counseling department in designing program and educational materials to strengthen social and family support is suggested. Nurses in psychiatric OPD may also collaborate with inpatient unit to assess patients’ supportive system, and connect with community health service in increasing patients’ social network, finally patients’ supportive system is to be expanded through continuity of care.

One limitation of the present study was the cross-sectional study design, which minimized the causal relationship between social support and symptomatic remission. The classification of symptomatic remission was not included in the time standard proposed by the RSWG. Although we used CGI-I to assess symptom stability within six months, only overall improvements were achieved rather than full remissions as defined by the RSWG. Second, previous studies have reported that the DUP of patients with mental illness is one of the most relevant clinical factors for the relief of schizophrenia symptoms. However, because the participants of the present study were mostly had chronic schizophrenia (for more than a decade), the time of the first episode and the DUP were untraceable. Although medical records were reviewed, not every primary diagnosis was recorded in the hospital or had a detailed record. We suggest that future studies of schizophrenia patients include the variable “DUP after treatment” to facilitate the validation of t the association between the variable and symptomatic remission. In terms of sample selection, all cases were collected from a medical center in northern Taiwan; therefore, caution should be exercised before the generalization of the results. In addition, the results of the present study were limited to the remission status of patients with chronic schizophrenia, because we included patients from a psychiatric OPD and day wards, and the number of outpatients was fivefold that of the day ward patients. Finally, considering the relatively small sample size, in the future, random sampling methods should be adopted to enroll much more participants across medical institutions in Taiwan to explore the remission rates of patients with chronic schizophrenia in communities.

## 5. Conclusions

The present study adopted the RSWG definition for symptomatic remission and applied the CGI-I to explore the remission rates of participants. Less than 50% of the schizophrenia patients living in the community achieved symptomatic remission. Social support, especially family support, was a factor that facilitated the achievement of symptomatic remission status in the schizophrenia patients. Therefore providing health education and rehabilitation information to family caregivers is suggested [6], so that the families can learn to manage patient symptoms effectively and obtain the access to resources that could assist patients, which, in turn, would enhance and reinforce family social support. In Asia, only few studies have explored symptom remission in patients at different stages of schizophrenia. Therefore, the comprehensive understanding of factors influencing remission rate is yet to be achieved. Greater application of symptomatic remission criteria is anticipated, and the association between more psychosocial factors such as depression, social function, and quality of life and remission status in the study population should be investigated. We also recommend the adoption of longitudinal research designs in future to enhance the explanatory power of causality.

## Figures and Tables

**Figure 1 ijerph-18-03977-f001:**
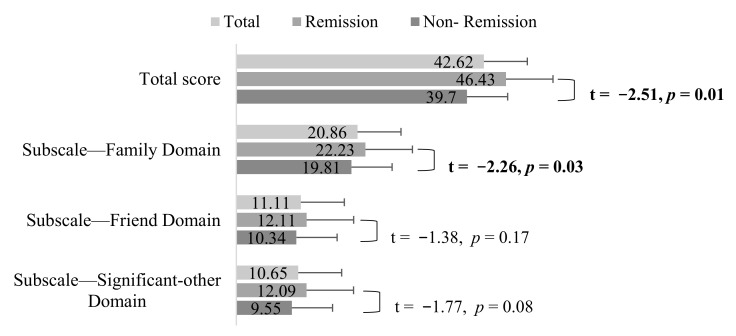
Comparison of social support between symptomatic remission and nonremission.

**Table 1 ijerph-18-03977-t001:** Comparison of socio-demographic characteristics and healthy lifestyles between symptomatic remission and non-remission.

Variable	Total	Remission	Non-Remission	$x^{2}/t$	*p*
	*n* (%)	*n* (%)	*n* (%)		
Total	129	56 (43.0)	73 (57.0)		
Gender (%)				0.11	0.73
Male	60 (46.5)	27 (45.0)	33 (55.0)		
Female	69 (53.5)	29 (42.0)	40 (58.0)		
Age (years)(Mean ± SD ^1^)		45.1 ± 12.3	45.7 ± 10.7	0.30	0.76
<40 (years)	45 (34.8)	24 (53.3)	21 (46.7)	2.77	0.09
≥40 (years)	84 (65.2)	32 (38.1)	52 (61.9)		
Marital status				0.06	0.79
Single (%)	107 (82.9)	47( 43.9)	60 (56.1)		
Married (%)	22 (17.1)	9 (40.9)	13 (59.1)		
Living status				1.38	0.23
Living Alone	6 (4.7)	4 (66.7)	2 (33.3)		
With Family	123 (95.3)	52 (42.3)	71 (57.7)		
Education status				2.21	0.33
Junior high or under	41 (32.0)	14 (34.1)	27 (65.9)		
Senior high	54 (42.0)	25 (46.3)	29 (53.7)		
University or above	34 (26.0)	17 (50.0)	17 (50.0)		
Occupation				0.21	0.64
Unemployed	80 (62.0)	36 (45.0)	44 (55.0)		
Employed	49 (38.0)	20 (40.8)	29 (59.2)		
Annual family income				0.13	0.71
≤ US $16,300	92 (71.3)	39 (42.4)	53 (57.6)		
> US $16,300	37 (28.7)	17 (45.9)	20 (54.1)		
Family History of mental disorders				3.47	0.06
No	106 (82.1)	42 (39.6)	64 (60.4)		
Yes	23 (17.9)	14 (60.9)	9 (39.1)		
Current smoke status				0.99	0.32
No	102 (79.0)	42 (41.2)	60 (58.8)		
Yes	27 (21.0)	14 (51.9)	13 (48.1)		
Exercise/Weekly				2.69	0.10
No	75 (58.1)	28 (37.3)	47 (62.7)		
Yes	54 (41.9)	28 (51.9)	26 (48.1)		
Adherence to antipsychotic medications				0.46	0.49
No	22 (17.0)	11 (50.0)	11 (50.0)		
Yes	107 (83.0)	45 (42.1)	62 (57.9)		

^1^ Standard deviation.

**Table 2 ijerph-18-03977-t002:** Comparison of clinical characteristics between symptomatic remission and non-remission.

Variable	Total	Remission	Non-Remission	x^2^/t	*p*
	*n* (%)	*n* (%)	*n* (%)		
Age of onset (years) (Mean ± SD ^1^)		32.1 ± 11.7	32.8 ± 12.1	0.34	0.73
≤18 (years)	12 (9.3)	7 (58.3)	5 (41.7)	1.21	0.55
18–40 (years)	84 (65.1)	35 (41.7)	49 (58.3)		
>40 (years)	33 (25.6)	14 (42.4)	19 (57.6)		
Duration of illness (years) (Mean ± SD)		12.9 ± 8.6	12.8 ± 7.7	−0.10	0.92
Duration of untreated illness within 1 year (days) (Mean ± SD)		6.5 ± 24.5	5.93 ± 27.4	−0.12	0.90
Current treatment				0.18	0.67
OPD	98 (75.9)	46 (42.6)	62 (57.4)		
Day care	21 (16.1)	10 (47.6)	11 (52.4)		
Previous psychiatric hospitalization				0.01	0.93
No	64 (49.6)	28 (43.8)	36 (56.3)		
Yes	65 (50.4)	28 (43.1)	37 (56.9)		
No. of hospitalization		1.55 ± 2.3	1.8 ± 2.4	0.65	0.51
never	64 (49.6)	28 (43.8)	36 (56.3)	0.83	0.65
1–3 times	40 (31.0)	19 (47.5)	21 (52.5)		
>3 times	25 (19.4)	9 (36.0)	16 (64.0)		
Last discharge				2.28	0.31
never	64 (49.6)	28 (43.7)	36 (56.3)		
Less than 6 months	11 (8.5)	7 (63.6)	4 (36.4)		
Over 6 months	54 (41.9)	21 (38.9)	33 (61.1)		
Antipsychotic treatment regiment				0.01	0.93
Oral only	102 (79.7)	44 (43.1)	58 (56.9)		
Combine with oral and long-acting injection	26 (20.3)	11 (42.3)	15 (57.7)		
Antipsychotic medication					0.35
Atypical-included	121 (94.5)	53 (43.8)	68 (56.2)		
Atypical-not included	7 (5.5)	2 (28.6)	5 (71.4)		
CGI-I (Mean ± SD)		3.0 ± 1.1	3.4 ± 0.8	2.51	0.013
≤4 (%)	125 (96.9)	56 (44.8)	69 (55.2)	3.16	0.07

^1^ Standard deviation.

**Table 3 ijerph-18-03977-t003:** Logistic regression of the association between social support and symptom remission.

Variable	B	SE ^1^	*p*-Value	AOR ^2^	95% CI ^3^	
Age (≥40 years vs. <40 years)	0.68	0.41	0.10	---	0.89	4.35
Family History of mental disorder (Yes vs. No)	0.68	0.51	0.18	1.98	0.73	5.35
Exercise/Weekly (Yes vs. No)	0.33	0.41	0.42	1.38	0.62	3.08
Social support-Family domain	0.07	0.03	0.04	1.07	1.01	1.14
Social support-Friend domain	−0.003	0.03	0.92	0.99	0.94	1.06
Social support-Significant other	0.03	0.03	0.25	1.03	0.98	1.09

^1^ Standard error, ^2^ Adjusted Odds Ratio, ^3^ Confidence intervals.

## Data Availability

The data presented in this study are available on request from the corresponding author. The data are not publicly available due to privacy restrictions.

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
