# Peer review of "The Association of Social Support and Symptomatic Remission among Community-Dwelling Schizophrenia Patients: A Cross-Sectional Study"

_ijerph, 2021, doi:10.3390/ijerph18083977_

Round 1

Reviewer 1 Report

In the present manuscript, the authors examined the association between family/social support and symptomatic remission outcomes in schizophrenia patients recruited from a psychiatric outpatient department and a psychiatric day care center in Taiwan. The Authors examined 129 patients using PANSS-8, CGI-I, and MSPSS questionnaires and collected clinical characteristics data by reviewing medical records. The Authors concluded that „the major finding of the study is that schizophrenia patients with higher social support levels experience higher rates of remission, and family support is an indispensable factor for symptomatic remission”.

The study is correctly designed, the methods used are adequate, the data are neatly presented, and the results support the conclusions. Moreover, the authors carefully analyzed the limitations of their study (Discussion, page 11, last paragraph).

Regrettably, despite all those advantages, I can not see the novelty of the study. I do not feel that the question raised in the article is original and provides an advance in current knowledge on the association between social support and illness remission. It is commonly accepted that family/social support is an indispensable component of the therapy of people with mental illness and may have a critical  role on the clinical and recovery outcomes of psychiatric patients. The weakest part of the manuscript is Discussion which brings nothing new and contains inconsistencies. For example, the Authors give cultural differences between the East and the West as the reason for the discrepancy between their and previously published results of  Hamaideh et al. (J. Psychiatr. Ment. Health Nurs 2014). However, the paper of Hamaideh concerns the Jordanian patients with schizophrenia, and in Jordanian culture and society, family (both immediate and extended) plays a central role alike in Taiwan. The next paragraph of the Discussion (pages 10-11, starting with the words „Another study on Chinese patients reported…”), although it raises important issues, is redundant because it is not about the subject of the study.

Reviewer 2 Report

The manuscript by Chi-Hsuan Fan and colleagues showed a cross-sectional study of the association of social support and symptomatic remission among community-dwelling schizophrenia patients. This manuscript is novel and well organized. However, I just have some minor concerns.

References missing-“Symptom remission and functional recovery are considered prognostic indicators in patients with schizophrenia.” “Certain sociodemographic characteristics are associated with remission status in schizophrenia patients.”

Please write the reference at the end of a sentence, so the reader can read uninterruptedly, for example- “Lang et al. [26] reviewed 21 studies and revealed a sustained remission rate of 52.6% after 6 months in some studies.” “According to Cohen and Wills [28], social support can be divided into es- teem support, informational support, social companionship, and instrumental support.” “In a previous study, early social support, 1 year after treatment, predicted remission in negative symptoms and functional recovery at 5 years in patients with psychotic disor- ders[34], and social support in the study was assessed by the case managers of patients based on three items from the Wisconsin Quality of Life Scale-Provider Version”

Representation of the data is thorough; however, I would recommend giving some bar graphs on the parameters of social support and remission status.

Reviewer 3 Report

This paper examined the relationships between social support and symptomatic remission among community-dwelling schizophrenia patients. This study is well worth publishing because the theoretical background and research design are well developed. The following points need to be considered.

First, IRB-related content should be described in the research content.

Second, explanations related to survey implementation, process, and sampling should be added.

Third, it is necessary to add practical implications, such as designing the policy program that can strengthen social support.

Round 2

Reviewer 1 Report

The authors provided comprehensive explanations and introduced appropriate corrections. I have no more comments.